The urinary bladder wall is remodeled by undulatory resistance training in female Wistar rats

Braverman Amyr 1
Dsouki Nuha A. 1
Veridiano Juliana M. 1
Paunksnis Marcos R. R. 2
http://orcid.org/0000-0002-9456-192X Maifrino Laura B. M. 3
Rica Roberta L. 4
Bocalini Danilo S. 2
Pereira Bruno F. 5
Pitol Dimitrius L. 6
Cafarchio Eduardo M. 1
http://orcid.org/0000-0002-7991-4117 Chess-Williams Russ 7
http://orcid.org/0000-0001-9970-817X Aronsson Patrik 8 patrik.aronsson@pharm.gu.se
http://orcid.org/0000-0001-8627-363X Sato Monica A. 1
1 Department of Morphology and Physiology, Centro Universitario FMABC , Santo Andre, SP , Brazil
2 Laboratory of Experimental Physiology and Biochemistry, Federal University of Espirito Santo , Vitoria, ES , Brazil
3 Dante Pazzanese Institute of Cardiology , Sao Paulo, SP , Brazil
4 Department of Physical Education, Faculty Estacio de Sa , Vitoria, ES , Brazil
5 Laboratory of Molecular and Translational Endocrinology, Dept. of Medicine, Federal University of Sao Paulo , Sao Paulo, SP , Brazil
6 Laboratory of Histotechnology, University of Sao Paulo , Ribeirao Preto, SP , Brazil
7 Centre for Urology Research, Bond University, Faculty of Health Science & Medicine , Gold Coast, QLD , Australia
8 Department of Pharmacology, University of Gothenburg , Gothenburg , Sweden
Barnett Matthew
Electronic publication date: 2025 Mar 31
Publication date: 2025
Volume: 13
Electronic Location ID: e19172
Received 2024 Sep 4; Accepted 2025 Feb 24
Copyright: © 2025 Braverman et al.
Copyright year: 2025
Copyright holder: Braverman et al.
License: This is an open access article distributed under the terms of the Creative Commons Attribution License, which permits unrestricted use, distribution, reproduction and adaptation in any medium and for any purpose provided that it is properly attributed. For attribution, the original author(s), title, publication source (PeerJ) and either DOI or URL of the article must be cited.
License URL: https://creativecommons.org/licenses/by/4.0/

Keywords: Undulatory resistance training, Urinary bladder, Collagen, Metalloproteinase, TIMP, Elastin, Picrosirius, Resorcin

Funding: PIBIC-CNPq Sao Paulo State Research Foundation FAPESP, grant#2018/00191-4 NEPAS-FMABC Centro Universitario FMABC This work was supported by PIBIC-CNPq, and Sao Paulo State Research Foundation (FAPESP, grant#2018/00191-4), NEPAS-FMABC, and Centro Universitario FMABC. The funders had no role in study design, data collection and analysis, decision to publish, or preparation of the manuscript.

==============================
Urinary stress incontinence has a high prevalence in women, with many associated risk factors, such as high impact and intensity sports due to increased intra-abdominal pressure causing stretching and weakening of the pelvic floor muscles. No previous study has investigated the effects of undulatory resistance training (URT), deemed as high impact sports’s modality, on urinary bladder (UB) and tissue remodeling. Healing of tissue depends on the equilibrium of metalloproteinases (MMPs) and their inhibitors (TIMPS). We aimed to investigate the histomorphological effects of URT on UB wall. Twelve female Wistar rats were randomly divided in two groups: sedentary (SED, n = 5) and URT (n = 7). URT was performed with a ladder climbing equipment after the maximum loaded carrying test (MLCT) was carried out. The training sessions were organized in three blocks increasing the MLCT’s weight each block. New MLCT were set at the end of each block. The day after the last training, the rat was euthanized and the UB was harvested and stored in formalin for later histological analysis stained with hematoxylin-eosin (HE), Masson’s trichrome (MT), picrosirius-hematoxylin (PH) and resorcin-fuchsin (RF), and immunohistochemistry for metalloproteinase-1 (MMP1) and tissue inhibitor of metalloproteinase (TIMP1). UB slices of URT rats stained with HE showed changes in all UB layers, with increased thickeness of the urothelium. MT staining allowed to observe an increased collagen concentration on the lamina propria layer (LP) of URT rats. PH staining demonstrated a higher luminous intensity for collagen type I and III in lamina propria and smooth muscle layers of the UB wall in the URT group than in SED. RF staining demonstrated an increase of elastic fiber concentration on the LP and smooth muscle layer of the bladder wall in the URT group. Immunohistochemistry of UB slices showed that MMP1 and TIMP1 were immunolabeled on the LP the UB wall in URT rats, with TIMP1 showing a lighter labeling than MMP1. Therefore, the findings suggest that URT induces remodeling of the urinary bladder wall characterized by imbalance between MMP1 and TIMP1 and evoking an alteration in the connective tissue from loose to dense.

Introduction

The urinary bladder (UB) composes the urinary system together the kidneys, ureters and urethra (Amerman, 2018). The UB is organized in following histological layers: lining epithelium (urothelium), lamina propria (a suburothelial layer separating the urothelium from the underlying muscle, consisting of connective tissue), muscularis propria (detrusor muscle) and serosa/adventitia (Bolla et al., 2023).

The UB wall and its internal sphincter are innervated by the autonomic nervous system (ANS). The micturition reflex depends on neural mechanisms, which control the coordination of the detrusor muscle of the UB and the urethra components. The parasympathetic branch of the ANS emerges from the sacral region of the spinal cord and is responsible for UB contraction, whereas the sympathetic efferents originate from the lumbar-thoracic region of the spinal cord and are responsible for UB relaxation (Andersson & Hedlund, 2002; de Groat, Griffiths & Yoshimura, 2015). Stretch and nociceptive receptors initiate the micturition reflex by signaling through afferent fibers, which send the information to the spinal cord. Then, parasympathetic efferent fibers are activated and evoke the micturition reflex. Nevertheless, this reflex arch might be influenced directly by facilitatory or inhibitory brain centers (Yoshimura & de Groat, 1997; de Groat, Griffiths & Yoshimura, 2015).

The onset of the micturition reflex is facilitated by the pontine micturition centre (PMC) also known as Barrington’s nucleus, whereas the pontine storage centre (PSC) stimulates urinary storage. Voluntary control, emerging from the cerebral cortex, acts as activator or deactivator of these pontine centers. The cortical influence can stimulate the PMC and activates the parasympathetic efferents on the UB, allowing the micturition reflex (de Groat, 1998; Yoshimura & de Groat, 1997; Sugaya et al., 2005).

There are several risk factors for stress urinary incontinence (SUI), such as pregnancy, labor, pelvic prolapse organs, congenic anomalies, perimenopausal estrogen deficiency, injuries or surgeries on pelvic organs (Hunskaar et al., 2000); as well as intense physical stress and professional sporting (Poświata, Socha & Opara, 2014; Caetano, da Consolação Gomes Cunha Fernandes Tavares & de Moraes Lopes, 2007; Bump & Norton, 1998). Furthermore, SUI has been frequently observed on volleyball and hockey athletes and its prevalence rises almost 25% in nulliparous young athletes (Greydanus, Omar & Pratt, 2010; Bø & Borgen, 2001). Another study with 156 young nulliparous athletes has shown that 28% of them present SUI symptomatology while practicing the sport. Those who practice high impact exercises with jumpings, high impact landing or running were often affected by the disorder (Nygaard, Glowacki & Saltzman, 1996). Since the pelvic floor muscles are responsible for the mechanic area stabilization (pelvic and spinal), the current hypothesis is that high impact exercises cause stretching and weakening of pelvic floor muscles by pressure overload activity (Caetano, da Consolação Gomes Cunha Fernandes Tavares & de Moraes Lopes, 2007; Snijders, Vleeming & Stoeckart, 1993a, 1993b; Nygaard et al., 1990; Fozzatti et al., 2012; Bø, 2004; Jiang et al., 2004).

Exercise against resistance, regional muscular training with weight carrying and frequent repetitions, has primarily been used for muscle hypertrophy as well as for treatment of chronic diseases and healthy lifestyle (Bermudes et al., 2004; Pollock & Froelicher, 1990; Kelley, 1997). A study with cardiac rehabilitation patients shows an improvement on hemodynamic pattern in patients performing resistance exercise (Verrill & Ribisl, 1996). Different patterns of resistance training have been developed with periodical changes on volume and intensity of weight and/or repetitions. Commonly, the linear pattern (classic) and non-linear pattern (undulatory) are used as training program (Santos et al., 2014). The difference between those is the periodical change where the: “linear pattern” maintains the same exercise routine and weight for a determined time cycle, whereas the “undulatory pattern” changes weight and intensity daily or weekly (Santos et al., 2014; Fleck, 2011). Although both patterns elicit increases in strength, and induce metabolic and functional improvements, it has been suggested that the non-linear pattern of exercise shows better results, however, the mechanisms involved in this outcome are still not fully understood (Bø, 2004; Prestes et al., 2009; Ahmadizad et al., 2014; Foschini et al., 2010; Rhea et al., 2002; Strasser & Pesta, 2013; Cuff et al., 2003; Newton et al., 2002).

High-impact sports have been considered a risk factor for the development of urinary incontinence. Particularly, exercises that allow the feet to come into contact with the ground and consequently generate a reaction force, can increase by many times the body weight (Hay, 1993; Caetano, da Consolação Gomes Cunha Fernandes Tavares & de Moraes Lopes, 2007). This effect caused by impact exercises can affect the continence mechanism by altering the amount of force transmitted to the pelvic floor. This can contribute to urinary incontinence even among young nulliparous women as well as in practitioners of high-impact sports. Different types of exercise with periodization protocols can manipulate training variables to optimize muscle adaptation and force development. Linear periodization and the nonlinear or daily undulatory periodization training program have been widely used to improve muscle strength. Linear periodization is characterized by gradual increases in training intensity and decreases in volume (Rhea et al., 2002). A nonlinear periodization program is characterized by more frequent alterations in intensity and volume, performed on a weekly or daily basis (Poliquin, 1988). Due to differences in training characteristics between the linear and nonlinear periodization program (e.g., frequency and exercise order), differences in mechanical stresses can be imposed (Toigo & Boutellier, 2006; Ratamess et al., 2009), which in turn may result in differences in neuromuscular adaptation and subsequent force development. Exercises that demand a lot of physical effort and cause high impact yields an excessive increase in the intra-abdominal pressure. Thus, the pelvic organs would be overloaded by the abdominal region, and pushed down, damaging the muscles responsive for the support of these organs (Caetano, da Consolação Gomes Cunha Fernandes Tavares & de Moraes Lopes, 2007). However, Magaldi et al. (2020) have shown that climbing with 75% body weigh load for 3 weeks in female rats, with the loads increased gradually and characterizing a linear model of exercise, can induce a decrease in the smooth muscle layer of the urinary bladder wall compared to sedentary rats. In addition, the responsiveness of the urinary bladder to acetylcholine and noradrenaline, evaluated in vivo by changes in intravesical pressure, was markedly decreased in rats submitted to that approach of resistance exercise. Thus, it is already known that a linear model of exercise evokes changes in the urinary bladder wall and function. Nevertheless, it is unknown whether a undulatory resistance training can induce or not any remodeling in the bladder wall.

It is also unknown if only the stretching and weakening of pelvic floor muscles are responsible for SUI observed on athletes. Although SUI has multiple risks factors associated, no previous study has showed if URT could cause structural bladder damage and consequent changes in UB function. Thereby, in this study, we focused to investigate the effect of URT on the histomorphology of the UB layers stained by hematoxylin-eosin, as well as resulting changes in the structure of the extracellular matrix stained with Masson’s trichrome, picrosirius-hematoxylin, or resorcin-fuchsin to verify possible modifications in the collagen and elastic fibers. Using an immunohistochemistry assay, we also evaluated the presence of metalloproteinase-1 (MMP1, an interstitial collagenase) and the tissue inhibitor of metalloproteinase (TIMP1, and inhibitor of collagenase) in the extracellular matrix of the UB wall in rats submitted to URT.

Materials and Methods

Animals

The Animal Ethics Committee of Centro Universitario FMABC provided full approval of this study (document approval number 03/2020). We used adult female Wistar rats (~250 g, 12 weeks-old) supplied by the Animals Care of Centro Universitario FMABC. Rigid environment control rules were applied to minimize bias: four animals were housed in each plastic cage, and the stainless-steel cage lid was high enough to allow the animals for standing and provide sufficient space for play. Rats also had the opportunity for activity as tugging or gnawing paper material through the cage lid. A 12:12 h light-dark cycle; humidity ~70% and room temperature at ~23° Celsius was maintained in the animal’s room. The animals had water and standard chow pellets (Nuvilab®, Seoul, South Korea) available ad libitum. A total of 12 animals were used in this study. The sample size was calculated using the resource equation method (Festing & Altman, 2002; Festing, 2006; Charan & Kantharia, 2013).

Experimental design

Rats were randomly divided in two groups: Sedentary (SED, control, n = 5) and Undulatory Resistance Training (URT, n = 7). This study had no blinding, all staff and researchers knew each rats’ specifications. The URT group was submitted to the undulatory resistance training protocol. One day after the last exercise bout, rats of all groups were euthanized with an overdose of sodium thiopental (100 mg/kg, i.p.), submitted to laparotomy and the UB was harvested and stored in 10% formalin solution. All the animals were euthanized at the planned end time point and none animal survived at the end of the experiment. Afterwards, the UB was embedded in paraffin and the tissue were sliced in a microtome (Lupetec®, Boston, MA, USA) and used for staining with hematoxylin-eosin, Masson’s Trichrome, picrosirius-hematoxylin, resorcin-fuchsin as well as for immunohistochemistry for MMP1 and TIMP1.

Undulatory resistance training protocol

1st Phase: Familiarization to the ladder (height: 110 cm and 80° angle from floor) and to the rest box (20 × 20 × 20 cm) at the top of the ladder for 5 days. In this phase, each rat climbed 3 times per day starting in different positions of the ladder (upper third, middle and base), with 60 s of resting between each climbing.

2nd Phase: Maximum load carrying test (MLCT) was carried out in the day six of the protocol. All rats received loads in their tails that were gradually increased before climbing the ladder. 1st climb: 50% of body weight (b.w.) load, 2nd climb: 75% of b.w. load, 3rd climb: 90% of b.w. load, 4th climb: 100% of b.w. load, next climbs: 30 g was added to the load in the tail until climbing failure.

3rd Phase: Daily URT consisting of three blocks of training bouts for five consecutive days of training per week for 4 weeks. Each block had loads and intensity (climbing sets) values modified. 1st block: 60% of MLCT weight and 15 sets/day; 2nd block: 75% of MLCT weight and 12 sets/day; 3rd block: 90% of MLCT weight and 10 sets/day. Resting time after each climbing was 90 s in all blocks. A new MLCT was carried out at the end of each block. The sedentary (SED) group was taken to the training room to maintain the same environmental conditions.

The timeline of the URT protocol is depicted in Fig. 1.

Figure 1 Timeline of the undulatory resistance training (URT) protocol.

MCLT, maximum load carrying test; b.w., body weight; UB, urinary bladder.

Staining and analyses method

Hematoxylin-eosin staining: first, the slices were immersed in xylol and hydrated with degraded alcohol sequentially (100°, 90° and 70° GL) and immersed in distilled water for 1 min. Afterwards, the slices were stained with hematoxylin solution for 5 min, washed in running water for 4 min and stained with eosin solution for 1 min. The sections were dehydrated through an ascending ethanol series (70°, 90° and 100° GL), cleared in xylene, and mounted with cover slips glued with Entellan (Merck®, Rahway, NJ, USA). Paulete & Beçak (1976), described that hematoxylin has a basophile affinity and stains cellular nuclei in blue, whereas eosin has an acid affinity and stains cytoplasm in pink. For stereological analyses a light field microscopy was used (Nikon, Eclipse E-200, Tokyo, Japan). The layers were measured using ImageJ 3.0 software ruler (NIH, Bethesda, ML, USA). All measurements performed on UB wall followed a standard 100× amplification. Six slices of the bladder for each rat were evaluated. Each bladder slice was divided in four fields (0°–90°, 90°–180°, 180°–270°, 270°–360°) and the measurements were carried out using the ImageJ 3.0 software. Thus, each bladder was subjected to four measurements per field on each bladder section. All data presented in this study shows the average of the measurements/layer. Nevertheless, in order to calculate the ratios for the urothelium, lamina propria, and smooth muscle layer thicknesses relative to total UB wall thickness, the ratio for each individual sample was first calculated and then averaging those ratios across the group.

Masson’s Trichrome staining: EasyPath® kit (EP-11-20013) solutions were used for this staining. First, slices were immersed in xylol and hydrated with degraded alcohol sequentially (100°, 90° and 70° GL), after that, immersed in distilled water for 1 min, then stained with Weigert’s ferric hematoxylin for 10 min, and washed in running water for 2 min. On the next step, fuchsin was added for 5 min on the slices followed by immersion in phosphomolybdic acid solution for another 5 min, and immediate submersion into methyl blue solution for 5 min. Finally, the sections were dehydrated through an ascending ethanol series (70°, 90° and 100° GL), cleared in xylene, and mounted with cover slips glued with Entellan (Merck®, Rahway, NJ, USA). As previous described (Lillie, 1976), Masson’s Trichrome is a staining technique useful for differentiation of collagen fibers (blue stain) and muscle fibers (red stain). For qualitative analyses of collagen concentration, a light field microscopy was used (Nikon, Eclipse E-200, Tokyo, Japan).

Picrosirius-hematoxylin staining: Picrosirius EasyPath® (code EP-11-20011) kit solutions were used for this staining. First, the slices were immersed in xylol and hydrated with degraded alcohol sequentially (100°, 90° and 70° GL) and washed in running water for 1 min. Afterwards, slices were stained with Picrosirius solution for 1 h followed by washing in running water. Following that, hematoxylin solution was applied for 4 min, and the slices were washed in running water for 5 min. Finally, the sections were dehydrated through an ascending ethanol series (70°, 90° and 100° GL), cleared in xylene, and mounted with cover slips glued with Entellan (Merck®, Rahway, NJ, USA). As previously described (Kim et al., 2000), this staining technique provides the differentiation of type I (red stain) and III (green stain) collagen fibers. For qualitative analyses of collagen concentration, a polarized light field microscopy was used (Nikon, Eclipse E-200, Tokyo, Japan).

Resorcin-fuchsin staining: this staining technique is used for identification of elastic fibers (dark blue/black color). First, the slices were immersed in xylol and hydrated with degraded alcohol sequentially (100°, 90° and 70° GL) and washed in running water for 1 min. Afterwards, the slices were maintained in a resorcin-fuchsin solution for 1 h, washed in running water and immersed in alcohol 70° and washed again. Finally, the sections were dehydrated through an ascending ethanol series (70°, 90° and 100° GL), cleared in xylene, and mounted with cover slips glued with Entellan (Merck®, Rahway, NJ, USA). For qualitative analyses of collagen concentration, a polarized light field microscopy was used (Nikon, Eclipse E-200, Tokyo, Japan).

Immunohistochemistry assay for MMP1 and TIMP1 in the urinary bladder wall

MMP1 immunohistochemistry: UB samples preserved in paraffin were cut in a microtome (Lupetec®, Boston, MA, USA) with 5 µm thickness and the slices were fixated on histological glass slides. The slices were deparaffinized with xylene and rehydrated into graded alcohol sequency (100°, 90° and 70° GL) and washed in running water for 1 min. Antigen retrieval was enhanced by submerging slides in sodium citrate buffer (pH 6.0) for 3 min. Endogenous peroxidase activity was quenched by incubation in hydrogen peroxide for 40 min. Nonspecific immunoglobulin binding was blocked using 2% milk solution (Molico®, Redwood City, CA, USA) diluted in phosphate buffer solution (PBS) 1X for 60 min. The slices were then incubated with primary antibody anti-MMP1 (Cloud-Clone®, Katy, TX, USA) at a dilution of 1:500 on 2% milk solution (Molico®, Redwood City, CA, USA) diluted in PBS 1X for 2 h at room temperature. The slides were washed 3 times in PBS 1X and then further incubated with a biotinylated secondary antibody (Rabbit IgG antibody HRP, Santa Cruz Biotechnology, Dallas, TX, USA) for 40 min at room temperature. Antigen-antibody complexes were detected by avidin-biotin-peroxidase method using diaminobenzidine (Dako®, Glostrup, Denmark) as a chromogenic substrate. Finally, the slides were counterstained with hematoxylin, mounted with cover slips glued with Entellan (Merck®, Rahway, NJ, USA), and then examined in light field microscopy (Nikon, Eclipse E-200, Tokyo, Japan).

TIMP1 immunohistochemistry: The UB slices obtained as reported above for MMP1 immunohistochemistry protocol were firstly submitted to enhancement of the antigen retrieval by emerging slides in sodium citrate buffer (pH 6.0) for 3 min. Endogenous peroxidase activity was quenched by incubation in hydrogen peroxide for 40 min. Nonspecific immunoglobulin binding was blocked using 2% milk solution (Molico®, Redwood City, CA, USA) diluted in phosphate buffer solution (PBS) 1X for 60 min. The slices were incubated at 4 °C overnight with primary antibody anti-TIMP1 (SAB4502971®; Sigma-Aldrich, St. Louis, MO, USA) at a dilution of 1:50 on 2% milk solution (Molico®, Redwood City, CA, USA) diluted in phosphate buffer solution (PBS) 1X. Slides were washed three times in PBS 1X and then further incubated with a biotinylated secondary antibody (Rabbit IgG antibody HRP, Santa Cruz Biotechnology, Dallas, TX, USA) for 40 min at room temperature. Antigen-antibody complexes were detected by DAP substrate. The slides were counterstained with hematoxylin and then examined by light field microscopy (Nikon, Eclipse E-200, Tokyo, Japan).

Quantitative analysis of collagen, elastin, MMP1 and TIMP1 in the urinary bladder slices of URT and SED rats

The quantitative analysis of urinary bladder histological sections stained by Masson Trichrome and Resorcin-Fuchsin allowed the evaluation of collagen and elastin in the urinary bladder parenchyma. The urinary bladder slices immunostained using anti-MMP1 and anti-TIMP1 were also quantified to investigate the amount of MMP1 and TIMP1. The images were captured in a light field microscope Nikon Eclipse E-200® (Tokyo, Japan) coupled to a HD camera (Moticam 1080 Images Plus 3.0, 1920 X 1080 resolution) with 100× magnification. All the images were quantified using the Image J v. 1.5 4k software (NIH, Bethesda, ML, USA) using the plugin Color Deconvolution 2 needed for this achievement.

Statistical analysis

Measurements of the bladder layers were submitted to the Kolmogorov-Smirnov test for normality. Once the data was defined as parametric, results were expressed as mean ± standard error of the mean (SEM) and submitted to unpaired Student’s t-test using the software Prisma Graph Pad 10 for comparison between URT and SED rats. The significance level was set at p < 0.05. Data considered outliers were obtained by calculating upper boundary and lower boundary by taking two standard deviations from the mean of the values.

Results

Haematoxylin-eosin staining and morphometric analysis of the urinary bladder layers

As shown in Figs. 2A, 2B, 3A and 3D, and Table 1, we observed that the urothelium of the UB in rats submitted to URT (n = 7) had a predominance of globose stratified epithelium and was significantly thicker (+35.25%, p = 0.009) compared to SED rats (n = 5). Although the lamina propria layer of the UB in all rats of the URT group seemed to have a reduced thickness (−11.36%, Table 1) in comparison to SED rats, this difference was not statistically significant (p = 0.240). In comparison to the SED group, the URT group showed dense connective tissue (Figs. 2A, 2B, and 3A and 3D). In the smooth muscle layer of the UB, we observed that URT rats showed larger cell spacing and lighter staining for connective tissue, nevertheless, the thickness of the smooth muscle layer in URT rats was not statistically different (+2.52%, p = 0.854) from SED rats (Figs. 2A–2B, 3A and 3D and Table 1). The total UB wall thickness was not different in URT rats (8,272.95 ± 457.61 µm) compared to the SED group (8,381.15 ± 665.96 µm) (p = 0.897, Table 1). We also observed that the ratio of the urothelium layer per total UB wall thickness (0.048 ± 0.004) was significantly increased compared to the SED group (0.035 ± 0.004, p = 0.044). However, no significant difference was found neither in the ratio of the lamina propria layer per total UB wall thickness (0.320 ± 0.018 URT vs. 0.357 ± 0.024 SED, p = 0.228) nor in the ratio of the smooth muscle layer per total UB wall thickness (0.632 ± 0.017 URT vs. 0.608 ± 0.025 SED, p = 0.410).

Figure 2 Photomicrographs showing the urinary bladder wall slices of rats stained with hematoxylin-eosin, Masson’s trichrome, picrosirius—hematoxylin, picrosirius—hematoxylin polarized, resorcin-fuchsin in sedentary (SED, images A, C, E, G, I, respectively) or undulatory resistance trained (URT, images B, D, F, H, J, respectively) animals in 20× amplification.

URO, Urothelium layer; LP, Lamina propria layer; SM, smooth muscle layer.

Figure 3 Photomicrographs showing the urinary bladder wall of rats stained with hematoxylin-eosin, Masson’s trichrome, and picrosirius-hematoxylin polarized in sedentary (SED, images A, B, C, respectively) or undulatory resistance trained (URT, images D, E, F, respectively) animals in 100× amplification.

URO, Urothelium layer; LP, Lamina propria layer; SM, smooth muscle layer.

Table 1 Measurements of urinary bladder layers (100× amplification) in rats maintained sedentary (SED) or submitted to undulatory resistance training (URT).

Layer	SED (µm)	URT (µm)	%Δ URT/SED((URT-SED)/SED * 100)	Observations of changes in URT compared to SED	
Urothelium	291.86 ± 14.04	394.73 ± 27.63*	+35.25%	• Predominance of the globose stratified form in URT	
Lamina propria	2,990.01 ± 201.10	2,650.30 ± 180.88	−11.36%	• Dense connective tissue in URT compared to loose connective tissue in SED	
Smooth Muscle	5,099.29 ± 563.42	5,227.91 ± 373.04	+ 2.52%	• Lack of connective tissue between cells in URT	
Total UB wall measurement	8,381.15 ± 665.96	8,272.95 ± 457.61	−1.29%	NS	
Notes:

Data are as mean ± S.E.M.

* Different from SED (p < 0.05, unpaired Student’s t-test).

NS, Not significant.

Masson’s Trichrome staining and collagen area measurement in the urinary bladder wall

Masson’s Trichrome staining performed on the UB slices showed that rats submitted to URT presented a more intense violet/blue color in the lamina propria layer in comparison to SED rats (Figs. 2C, 2D and 3B and 3E), demonstrating a significant larger collagen concentration/amount of fibrosis in that area in URT (161.1 ± 6.1 pixels/µm2, n = 8) than is SED animals (125.7 ± 7.3 pixels/µm2, n = 4, p = 0.0056) (Fig. 4A).

Figure 4 Quantitative analysis of fibrosis, elastin, MMP1 and TIMP1 in urinary bladder slices of SED and URT rats.

(A) Amount of fibrosis (pixels/µm2) in the urinary bladder slices stained by Masson Trichrome in sedentary rats (SED, n = 5) or submitted to undulatory resistance training (URT, n = 8), (B) Amount of elastin (pixels/µm2) in the urinary bladder slices stained by resorcin-fuchsin in SED (n = 6) or submitted to URT (n = 3) rats, (C) Amount of MMP1 (pixels/µm2) in sedentary rats (SED, n = 4) or submitted to URT (n = 7), and (D) Amount of TIMP1 (pixels/µm2) in sedentary animals (SED, n = 4) or submitted to URT (n = 6). *P < 0.05 vs. SED (unpaired Student’s t-test).

Picrosirius Red staining and collagen fibers area measurements in the urinary bladder wall

Picrosirius Red staining demonstrated a higher yellow/red luminous intensity (collagen type I) as well as a higher green light color (collagen type III—elastic fiber) in the lamina propria and smooth muscle layers of the UB wall in the URT group than in SED rats (Figs. 2G, 2H, and 3C and 3F).

Resorcin-fuchsin staining in the urinary bladder wall.

Resorcin-fuchsin staining demonstrated the existence of elastic fibers in dark blue/black color present particularly in the lamina propria and smooth muscle layers of the UB wall, which were more intensively stained in the URT group than in SED rats (Figs. 1I–1J). URT rats presented a significant greater amount of elastin (21.4 ± 1.5 pixels/µm2, n = 6) than SED animals (13.0 ± 3.3 pixels/µm2, n = 3, p = 0.0284) (Fig. 4B). Collagen pink fibers of the smooth muscle layer were also stained more strongly in URT rats than in the SED group (Figs. 2I–2J).

Immunohistochemistry for MMP1 and TIMP1 in the urinary bladder wall

The immunohistochemistry assay showed that MMP1 was intensively labeled (golden/brown color) in the UB slices of rats submitted to URT. The majority of MMP1 labeling was observed in the lamina propria layer, particularly surrounding the smooth muscle in comparison to SED rats. Only a faint labeling for MMP1 was found on the UB slices in SED rats (Figs. 5A, 5B, 6A, 6B). The URT rats showed a significantly increased amount of MMP1 labeling (62.8 ± 3.3 pixels/µm2, n = 7) compared to SED animals (19.3 ± 7.1 pixels/µm2, n = 4, p = 0.0001, Fig. 4C).

Figure 5 Photomicrographs showing the immunohistochemistry for metalloproteinase-1 (MMP1, black arrows) on the urinary bladder wall of (A) sedentary rats (SED) or submitted to (B) undulatory resistance training (URT). The golden/brown color shows MMP1 immunolabeling. (C and D) Shows the photomicrographs with the immunohistochemistry for Tissue Inhibitor of Metalloproteinase-1 (TIMP1, blue arrows) on the urinary bladder wall of sedentary rats (SED) or submitted to undulatory resistance training (URT), respectively.

Amplification: 20×. URO, Urothelium layer; LP, Lamina propria layer; SM, smooth muscle layer.

Figure 6 Photomicrographs showing the immunohistochemistry for metalloproteinase-1 (MMP1, black arrows) on the urinary bladder wall of (A) sedentary rats (SED) or submitted to (B) undulatory resistance training (URT). The golden/brown color shows MMP1 immunolabeling. (C and D) Shows the photomicrographs with the immunohistochemistry for Tissue Inhibitor of Metalloproteinase-1 (TIMP1, blue arrows) on the urinary bladder wall of sedentary rats (SED) or submitted to undulatory resistance training (URT), respectively.

Amplification: 100×. URO, urothelium layer; LP, Lamina propria layer; SM, smooth muscle layer.

TIMP1 was immunolabeled with golden/brown color in the lamina propria layer of URT rats, whereas SED group only showed a spread weak labeling in the UB wall without layer predominance (Figs. 5C, 5D and 6C, 6D). Nevertheless, no difference was observed in the percent area of TIMP1 labeling comparing URT (38.9 ± 4.0, n = 6) and SED animals (42.68 ± 4.0, n = 4, p = 0.8161) (Fig. 4D).

Discussion

Our histomorphology analysis demonstrated that the undulatory resistance training evoked changes in all layers of the urinary bladder wall. The urothelium thickness increased in URT rats, and consequently the ratio between the urothelium thickness per total urinary bladder wall measurement was greater than in SED rats. In addition, the pattern of cells in the urothelium layer changed and mainly globose stratified epithelium was observed using the hematoxylin-eosin staining. Little is known about the direct effects of physical exercise on urothelium cells and its specific pathways, but changes have previously been described. Blacklock (1977) showed cytoscopic alterations on the urothelium of eight marathon runners with haematuria, in which previous urinary pathology had not been identified. Nevertheless, roughly 48 h after the running, localized contusions with loss of urothelium and fibrinous exudate were found at specific sites within the bladder. It is worth noting that the urothelium is a highly specialized epithelium lining in the lower urinary tract with a variable number (3 to 6 layers) of cells in humans (Jost, Gosling & Dixon, 1989), and three cell layers in rats (Cohen, 2013). Distinct cell types are present in the urothelium, with the basal cell layer composed of cuboidal cells resting on a basement membrane attached via hemidesmosomes. Intermediate cells tend to be larger, and if the urothelium has more than three cell layers, this is due to multiple layers of intermediate cells (Cohen, 2013). Despite the urothelium being mitotically quiescent, with a very low constitutive rate of cell turnover, it has a high regenerative capacity in response to damage (Varley et al., 2005; Baker & Suthgate, 2011). However, the urothelium frequently undergoes squamous metaplasia, most likely in response to chronic inflammatory stimuli (Cote, Mitra & Amin, 2009). The larger urothelium thickness observed after exercise in the present study might be related to a chronic inflammatory damage of the bladder evoked by URT. However, we could not measure inflammatory biomarkers throughout the training period at different time points, which is a limitation of this study and will require further investigation in future studies.

Lamina propria and smooth muscle layer morphometric analyses showed no significant difference comparing URT and SED groups. In contrast, the histomorphological analysis demonstrated differences between URT and SED rats. In SED animals, the original lamina propria (LP) showed loose connective tissue within fine collagen fibers, but high density of cells, blood vessels, and ground substance. URT rats presented a dense connective tissue within low density of cells and ground substance. These alterations suggest that a remodeling of the connective tissue was ongoing as a result of URT.

Fibroblasts are connective cells responsible for producing all extracellular matrix molecules, such as fibrillar components (collagen and reticular fibers) and ground substance components (hydrophilic glycosaminoglycans, proteoglycan and glycoproteins) responsible for tensile force to matrix (Junqueira & Carneiro, 2008). Furthermore, fibroblasts have mechanoreceptors on their plasmatic membrane, thus mechanical and tension stimuli activate the actin and fibroblast’s cytoskeleton proteins in order to organize and adapt the stretched fibers through fibroblast proliferation and activation, as well as intense production of extracellular macromolecules, particularly collagen fibers. When the tension stimulus ends, actins disorganize and cease this remodeling process (Nusgens et al., 1984). This fact underpins the findings of our study, in which the URT group undergoes a constant mechanical stimulus evoked by muscle contraction on the pelvis that activates fibroblast mechanoreceptors in the bladder wall to trigger remodeling process for replacing the connective tissue from loose to dense.

Elastic fibers underwent interesting alterations in the URT group. The density of the elastic fibers seemed to be increased, suggesting elastic fiber synthesis, a process which is otherwise predominant during the fetal development. In adult tissue, the perpetuation of elastic fibers relies more on maintenance and remodulations of pre-existing fibers to guarantee elasticity and adequate tissue function than synthesis of new fibers (Halper & Kjaer, 2014). This finding supports the inflammatory remodeling hypothesis. After tissue injury, the inflammatory process starts, composed of several overlapping phases. The first phase is inflammation where coagulation, platelet aggregation and degranulation of tissue factors occurs, resulting in substances like platelet-derived growth factor (PDGF), insulin-like growth factor1 (IGF-1) and others substances attracting inflammatory cells, such as neutrophils, to the affected area. Those cells secrete elastase and collagenase responsible for extracellular matrix degradation and, synchronically, secrete interleukin-1 (IL-1) and tumor necrosis factor (TNF-alfa) responsible for neovascularization, and macrophage chemotaxis through interleukin-8 (IL-8). In addition, macrophages secrete tissue factors, such as IL-1, IL-8 and TNF-alfa. Through positive-feedback, IL-1 stimulates fibroblasts to produce proteins responsible for degrading the extracellular matrix, such as metalloproteinases (MMP). The second phase is proliferation after the inflammatory phase ends in a negative-feedback loop, fibroblasts and endothelial cells become the main cells responsible for neovascularization and matrix proliferation. In this stage, elastic fibers might have a small synthesis following collagen III synthesis. Finally, the remodeling stage is characterized by the equilibrium between extracellular matrix protein synthesis and degradation (Tarnuzzer & Schultz, 1996; Nagase & Woessner, 1999). However, further studies with specific technique are still required to elucidate the elastic fiber synthesis.

Different metalloproteinases (MMP) participate in tissue remodeling. The collagenases (MMP1) are responsible for collagen type 1 cleavage, the most common collagen in the skin. Gelatinases (MMP2 and MMP9) degrade collagen type I and IV. Strome lysin (MMP3) degrade collagen and proteoglycans. The MMPs activities are modulated by growth factors and Tissue Inhibitors of Metalloproteinases, such as TIMP1 and TIMP2. TIMP1 forms an inhibitory-complex with MMP1, MMP2 and MMP3 and up-regulates cellular growth by increasing cell activity (e.g., keratinocyte and fibroblasts), whereas TIMP2 inhibits MMP2, but it has been also implicated in the activation of MMP2 on latent form increasing tissue proliferation by up-regulation of other inflammatory cells. Overall, the human genome codes for four TIMPs (TIMP1 to 4) responsible for inhibiting MMPs activities, but also promotes anti-angiogenic activity, apoptosis regulation and neuroplasticity (Tarnuzzer & Schultz, 1996; Woessner, 1994; Bellayr, Mu & Li, 2009; Brew & Nagase, 2010; Moore et al., 2012). The imbalance between MMPs and TIMPs are associated with the etiopathology of inflammatory and proliferative illnesses, such as cardiovascular, pulmonary, rheumatologic diseases and tissue ulceration (Brew & Nagase, 2010; Moore et al., 2012; Spinale, 2002; Loftus & Thompson, 2002). In our study, immunohistochemistry for MMP1 and TIMP1 demonstrated increased MMP1 labeling in the lamina propria of URT rats compared to SED animals. In contrast, the lamina propria showed a lighter TIMP1 labeling than the MMP1. We hypothesized that the lamina propria was remodeling due to mechanical stimuli on fibroblasts evoked by URT. Likely, inflammatory processes were evoked by the intense exercise, which is supported by the findings that the increased urothelium thickness and great amoubt of elastic fibers, suggesting an imbalance in extracellular matrix synthesis and degradation. Earlier studies have also demonstrated that the presence of TIMP1 and MMP1 dysregulation can be found in pathophysiological conditions, leading to the development of several diseases, thus, this evidence strenghtens the findings of the current study (Moore et al., 2012; Spinale, 2002; Kim et al., 2000; Roten et al., 2000). In our study, the rats submitted to URT have shown an increased amount of fibrosis in the lamina propria of the urinary bladder demonstrated by Masson’s Trichrome staining and larger labeling of MMP1 in the same bladder layer than in SED animals, however, without differences in TIMP1 labeling between groups. Thus, we could not exclude the hypothesis that the fibrosis in the bladder layer associated with an imbalance in MMP1 and TIMP1 would be able to produce a bladder dysfunction in long-term.

In a different model of resistance exercise, Magaldi et al. (2020) have shown that climbing with 75% body weigh load for 3 weeks in female rats can induce a decrease in the smooth muscle layer of the urinary bladder wall compared to sedentary rats. In addition, the responsiveness of the urinary bladder to acetylcholine and noradrenaline, evaluated in vivo by changes in intravesical pressure, was markedly decreased in rats submitted to that approach of resistance exercise (Magaldi et al., 2020). We have not carried out the functional evaluation of the urinary bladder, which is a limitation of this study. Further investigation will be necessary to investigate the effects of URT on cystometric parameters and the responsiveness of the urinary bladder to the mediators of the autonomic nervous system as acetylcholine or noradrenaline or other neuromodulators, such as ATP. The functional assessment will be helpful for a better understanding of the urinary bladder performance or the possible presence of dysfunction in rats submitted to URT.

Previous studies have shown that physical activity elicits a decrease in circulating sex hormones as estrogen in women (Ennour-Idrissi, Maunsell & Diorio, 2015). Lower urinary tract symptoms in postmenopausal women are linked to decreased estrogen levels (Zhang et al., 2024). Evidence in mice has demonstrated that estrogen treatment improves bladder function and reduce collagen deposition in the bladder tissues (Zhang et al., 2024). Although we have not measured the estrogen levels in the female rats of the current study, if the URT would be able to reduce this circulating hormone, the urinary bladder cells could increase the collagen deposition leading to a remodeling of the bladder layers.

Stress urinary incontinence has multiples risk factors associated, among those, high impact sports and intense exercises (Poświata, Socha & Opara, 2014; Caetano, da Consolação Gomes Cunha Fernandes Tavares & de Moraes Lopes, 2007; Bump & Norton, 1998; Greydanus, Omar & Pratt, 2010; Bø & Borgen, 2001; Nygaard et al., 1994). Although the well accepted current hypothesis is that pressure overload causes stretching and weakening of pelvic floor muscles (Bø, 2004) resulting in the stress urinary incontinence, our study demonstrated an intense bladder remodeling, in which lamina propria alters from loose to dense connective tissue. This remodeling likely changes the defense and nutrition roles of the loose connective tissue to instead provide increased resistance and flexibility in the dense connective tissue. In spite of the effects on the urinary bladder evoked by URT have been observed in female rats, these findings demonstrated the ability of the urinary bladder remodeling. Additional studies assessing the urinary bladder function upon URT will be also crucial to elucidate the benefits or not of this approach of exercise training and consequently for triggering future translational studies in athletes or individuals undergoing resistance training.

Conclusions

Our findings suggest that undulatory resistance training causes a remodeling process in the urinary bladder, evoking an imbalance in extracellular matrix synthesis and degradation, leading to a modification of the connective tissue from loose to dense.

Supplemental Information

Supplemental Information 1 ARRIVE 2.0 checklist.

Supplemental Information 2 Raw data and Mean-SD-SEM.

Raw data with mean, standard deviation (SD) and standard error of mean (SEM).

Supplemental Information 3 Raw data: individual measures of urinary bladder layers.

Supplemental Information 4 Data: mean and SEM and % calculation—urinary bladder layers.

Supplemental Information 5 Quantitative analysis: Image J.

Additional Information and Declarations

Competing Interests

The authors declare that they have no competing interests.

Author Contributions

Amyr Braverman conceived and designed the experiments, performed the experiments, analyzed the data, prepared figures and/or tables, authored or reviewed drafts of the article, and approved the final draft.

Nuha A. Dsouki conceived and designed the experiments, performed the experiments, analyzed the data, prepared figures and/or tables, authored or reviewed drafts of the article, and approved the final draft.

Juliana M. Veridiano conceived and designed the experiments, performed the experiments, authored or reviewed drafts of the article, and approved the final draft.

Marcos R. R. Paunksnis conceived and designed the experiments, performed the experiments, authored or reviewed drafts of the article, and approved the final draft.

Laura B. M. Maifrino conceived and designed the experiments, performed the experiments, authored or reviewed drafts of the article, and approved the final draft.

Roberta L. Rica conceived and designed the experiments, performed the experiments, authored or reviewed drafts of the article, and approved the final draft.

Danilo S. Bocalini conceived and designed the experiments, performed the experiments, authored or reviewed drafts of the article, and approved the final draft.

Bruno F. Pereira conceived and designed the experiments, performed the experiments, analyzed the data, prepared figures and/or tables, authored or reviewed drafts of the article, and approved the final draft.

Dimitrius L. Pitol conceived and designed the experiments, performed the experiments, analyzed the data, prepared figures and/or tables, authored or reviewed drafts of the article, and approved the final draft.

Eduardo M. Cafarchio conceived and designed the experiments, performed the experiments, analyzed the data, prepared figures and/or tables, authored or reviewed drafts of the article, and approved the final draft.

Russ Chess-Williams conceived and designed the experiments, analyzed the data, authored or reviewed drafts of the article, and approved the final draft.

Patrik Aronsson conceived and designed the experiments, analyzed the data, prepared figures and/or tables, authored or reviewed drafts of the article, and approved the final draft.

Monica A. Sato conceived and designed the experiments, performed the experiments, analyzed the data, prepared figures and/or tables, authored or reviewed drafts of the article, and approved the final draft.

Animal Ethics

The following information was supplied relating to ethical approvals (i.e., approving body and any reference numbers):

Animal Ethics Committee of Centro Universitario FMABC provided full approval for this research (document approval number 03/2020).

Data Availability

The following information was supplied regarding data availability:

The raw data is available in the Supplemental Files.

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
