# Peer review of "The urinary bladder wall is remodeled by undulatory resistance training in female Wistar rats"

_PeerJ, doi:10.7717/peerj.19172_

## Round 0.1 · original submission · Minor Revisions

This manuscript describes a well-designed and carefully conducted study. Both reviewers have proposed Minor Revisions, and I agree with this assessment.
While the authors should address each of the comments made, I note that both reviewers have highlighted a lack of data relating to inflammatory markers, and that there is no functional bladder assessment. I suggest the authors ensure they adequately address these points in particular.

·

Basic reporting

The study investigates the effects of undulatory resistance training (URT) on the urinary bladder (UB) of female Wistar rats. This study focuses on how URT affects bladder tissue, specifically through the histomorphological changes and the balance of metalloproteinases (MMPs) and their inhibitors (TIMPs), which play a role in tissue remodeling. URT induces remodeling of the UB wall, leading to structural changes in the layers of the bladder. URT increased the thickness of the urothelium, as well as collagen concentration in the lamina propria and smooth muscle layers. There were changes in the extracellular matrix, such as an increase in collagen types I and III and elastic fibers. Immunohistochemical analysis revealed that URT led to an imbalance in the expression of MMP1 and TIMP1, with higher MMP1 activity compared to TIMP1. The authors conclude that URT causes significant remodeling of the urinary bladder wall, which could potentially impact bladder function. Specific comments:

Experimental design

no comment

Validity of the findings

no comment

Additional comments

1. The introduction presents an adequate background on urinary stress incontinence and the potential effects of high-impact sports. However, the justification for focusing specifically on undulatory resistance training (URT) and its effects on the urinary bladder remains unclear. Could the authors elaborate on why URT was chosen as the training method and how it differs from other high-impact activities in terms of its potential to affect urinary bladder remodeling?
2. The histological analysis seems thorough, with multiple staining techniques used. However, the use of qualitative language, such as "increased" or "more intense," without accompanying quantitative data, may weaken the conclusions. It would be beneficial to provide more quantitative data or ratios for collagen and elastic fiber concentration changes to support the visual findings from the different staining methods.
3. The authors hypothesize that chronic inflammatory damage might be linked to the increased urothelium thickness observed in URT rats. However, no inflammatory markers were measured. Would it be possible to include or reference future studies that evaluate inflammatory biomarkers to strengthen this claim?
4. The paper mentions that no functional evaluation of the bladder (e.g., cystometric analysis) was conducted, which limits the ability to correlate structural changes with functional outcomes. Would the authors consider including or recommending functional bladder assessments in future studies to better link the observed histological changes with urinary bladder performance or dysfunction?
5. The immunohistochemical analysis highlights an imbalance between MMP1 and TIMP1 expression, with more pronounced MMP1 activity in URT rats. Could the authors explore whether this imbalance is associated with pathological conditions like fibrosis, and if so, how this might relate to urinary bladder dysfunction in the long term?
6. The study uses Wistar rats as a model, and while the results are valuable, the application of these findings to human physiology, particularly in athletes or individuals undergoing resistance training, is not addressed. Could the authors briefly discuss the potential translational value of these findings to human urinary bladder remodeling, or suggest further steps that need to be taken to bridge this gap?
7. The discussion offers possible explanations for the structural changes observed, such as mechanical stimuli activating fibroblasts, but does not explore alternative or additional pathways that might be at play, such as hormonal influences or neural regulation of bladder function. Could the authors expand the discussion to include other potential mechanisms, such as hormonal changes (e.g., estrogen) that might also contribute to the bladder remodeling observed in high-impact activities?

·

Basic reporting

Histology images lack proper white balance, and the scale in Figure 3 should be revisited.

Experimental design

Sex of Animals: Clarify if there was any specific reason for using only female rats and not including males.

Age of Rats: Since collagen deposition and bladder changes may vary by age, specifying the exact age of the adult rats used would be useful.

Undulatory Resistance Training Protocol: While the paper describes the three phases of the training protocol (Familiarization, MLCT, and URT), it’s not entirely clear that the first two phases serve as one-time, preliminary steps. Including a visual timeline or flowchart summarizing the protocol phases, weekly structure, and session breakdown could significantly enhance reader understanding. A timeline graphic would clarify that the three blocks of climbing (60%, 75%, and 90% of MLCT) are performed in each session and outline when MLCT recalculations are conducted.

Connective Tissue Density: The study observes differences in connective tissue density between groups, noting denser tissue in the URT group. While these differences are visually apparent in the images, adding quantitative measurements would strengthen the objectivity of the findings.

Ratio Calculation: The method used to calculate the ratios for the urothelium, lamina propria, and smooth muscle layer thicknesses relative to total UB wall thickness could be more rigorously approached. Based on the table, it seems the ratios are calculated by dividing the mean thickness of each layer by the mean total wall thickness for each group. A more precise approach would involve calculating the ratio for each individual sample first and then averaging those ratios across the group. This method, rather than using overall group means, would provide a more accurate representation of the variability in the data and may yield different ratio values.

Quantitative Measurements for Connective Tissue: While differences in connective tissue density are visually noticeable, adding quantitative metrics such as pixel area or signal intensity analysis would enhance the objectivity of these observations.

Validity of the findings

The study links increased MMP-1 and TIMP-1 expression to an inflammatory response triggered by platelet involvement, but this hypothesis may overreach given the limitations of the current data. Without measurements of inflammatory markers or direct evidence of platelet activation, attributing these changes solely to inflammation is speculative. Multiple cell types—such as urothelial cells, fibroblasts, detrusor smooth muscle (DSM), and immune cells—could also contribute to MMP-1 and TIMP-1 secretion. While fibroblasts are indeed prevalent in the lamina propria, the observed changes might instead reflect a mechanosensory adaptation. For example, with increased exercise demands, the bladder may need to produce higher pressure for micturition, leading to DSM adaptations in response to heightened contractile demands. As my own argument for a mechanosensory-based response lacks full support from the data, so does the inflammatory hypothesis in its current form. Additionally, in the conclusion section, the authors mention that these changes 'could ultimately induce dysregulation on bladder functionality.' Since no functional studies were conducted, it would be more accurate to keep conclusions focused on structural changes alone.

Additional comments

Overall, your study presents interesting results. With adjustments to temper conclusions, focusing on structural findings rather than speculative changes, the study’s clarity and impact could be strengthened.

---

## Round 0.2 · accepted · Accept

Both reviewers note that the authors have addressed their previous comments thoroughly, that the manuscript is much improved, and that it is suitable for publication. I agree with that assessment.

·

Basic reporting

The study investigates the effects of undulatory resistance training (URT) on the urinary bladder (UB) of female Wistar rats. This study focuses on how URT affects bladder tissue, specifically through the histomorphological changes and the balance of metalloproteinases (MMPs) and their inhibitors (TIMPs), which play a role in tissue remodeling. URT induces remodeling of the UB wall, leading to structural changes in the layers of the bladder. URT increased the thickness of the urothelium, as well as collagen concentration in the lamina propria and smooth muscle layers. There were changes in the extracellular matrix, such as an increase in collagen types I and III and elastic fibers. Immunohistochemical analysis revealed that URT led to an imbalance in the expression of MMP1 and TIMP1, with higher MMP1 activity compared to TIMP1. The authors conclude that URT causes significant remodeling of the urinary bladder wall, which could potentially impact bladder function. The revision of the manuscript is much improved, no additional comments.

Experimental design

N/A

Validity of the findings

N/A

Additional comments

N/A

·

Basic reporting

The manuscript is written in clear, professional English and provides sufficient references to support the background. The overall structure follows a logical flow, with properly formatted figures and tables.

Experimental design

They used a straightforward two-group approach (Sedentary vs. URT) to investigate bladder remodeling. Multiple staining methods (e.g., H&E, Masson’s Trichrome) provided a view of tissue changes. The sample size was justified, and the protocols were detailed enough to allow replication.

Validity of the findings

The results align well with the hypothesis, showing consistent changes in bladder structure after URT. Quantitative analyses of collagen, elastin, MMP1, and TIMP1 strengthen the conclusions. While functional tests are missing, the morphological evidence supports the authors’ main claims.

Additional comments

Thank you for addressing my comments so thoroughly. The additional data and clarifications significantly strengthen the manuscript, especially the quantitative findings for collagen, elastin, MMP1, and TIMP1. I appreciate how you refined the methods and added a clearer timeline for the URT protocol, which makes the study more transparent. Overall, your revisions enhance both the clarity and the impact of your work.